# A Novel GPPAS Model: Guiding the Implementation of Antimicrobial Stewardship in Primary Care Utilising Collaboration between General Practitioners and Community Pharmacists

**DOI:** 10.3390/antibiotics11091158

**Published:** 2022-08-27

**Authors:** Sajal K. Saha, Karin Thursky, David C. M. Kong, Danielle Mazza

**Affiliations:** 1School of Medicine, Faculty of Health, Deakin University, Geelong, VIC 3220, Australia; 2Department of General Practice, The School of Public Health and Preventive Medicine, Monash University, Melbourne, VIC 3168, Australia; 3National Centre for Antimicrobial Stewardship (NCAS), Department of Infectious Diseases, Melbourne Medical School, University of Melbourne, Melbourne, VIC 3000, Australia; 4Public Health Unit, Geelong Centre for Emerging Infectious Disease, Barwon Health, Geelong, VIC 3220, Australia; 5Centre for Medicine Use and Safety, Monash University, 381 Royal Parade Parkville, Melbourne, VIC 3052, Australia; 6Pharmacy Department, Ballarat Health Services, Ballarat, VIC 3350, Australia

**Keywords:** antimicrobial stewardship, implementation model, GP-pharmacist collaboration, primary care

## Abstract

Interprofessional collaboration between general practitioners (GPs) and community pharmacists (CPs) is central to implement antimicrobial stewardship (AMS) programmes in primary care. This study aimed to design a GP/pharmacist antimicrobial stewardship (GPPAS) model for primary care in Australia. An exploratory study design was followed that included seven studies conducted from 2017 to 2021 for the development of the GPPAS model. We generated secondary and primary evidence through a systematic review, a scoping review, a rapid review, nationwide surveys of Australian GPs and CPs including qualitative components, and a pilot study of a GPPAS submodel. All study evidence was synthesised, reviewed, merged, and triangulated to design the prototype GPPAS model using a Systems Engineering Initiative for Patient Safety theoretical framework. The secondary evidence provided effective GPPAS interventions, and the primary evidence identified GP/CP interprofessional issues, challenges, and future needs for implementing GPPAS interventions. The framework of the GPPAS model informed five GPPAS implementation submodels to foster implementation of AMS education program, antimicrobial audits, diagnostic stewardship, delayed prescribing, and routine review of antimicrobial prescriptions, through improved GP–CP collaboration. The GPPAS model could be used globally as a guide for GPs and CPs to collaboratively optimise antimicrobial use in primary care. Implementation studies on the GPPAS model and submodels are required to integrate the GPPAS model into GP/pharmacist interprofessional care models in Australia for improving AMS in routine primary care.

## 1. Background

Growing antimicrobial resistance associated with antimicrobial use is an increasing threat to healthcare and the global community [1,2]. Primary care has received much attention due to the use, overuse, and inappropriate use of antimicrobials for conditions including those that are not recommended for antibiotic therapy [3]. Although primary care patient services vary significantly by country, these services are ubiquitously less compliant with the concept of team-based patient care involving antimicrobial use [4]. A culture of independent antimicrobial-use decisions by doctors, pharmacists, and patients has become normalised in society which has substantially influenced inappropriate use of antimicrobials. A health system structure that promotes a cultural transformation to team-based antimicrobial decisions and patient care is instrumental [5] for antimicrobial stewardship (AMS) in primary care. 

General practitioners (GPs) and community pharmacists (CPs) are the most important antimicrobial stewards in primary care. Their collaboration, co-ordination, and communication are central to implement AMS strategies and to optimise antimicrobial use in routine GP and pharmacy practices [6,7]. CPs represent the third largest healthcare provider group globally [8], and they are still underutilized as resources for implementing AMS activities. The World Health Organisation and the Royal Pharmaceutical Society have policy propositions in favour of extending the roles of CPs for implementing AMS programmes in primary care. However, the national AMS action plans of many countries do not include clearly defined or policy-guided roles for CPs, and where defined, these roles are less supported by the healthcare system [9,10].

Growing evidence [11,12] corroborates that extending the roles of CPs beyond traditional dispensing is associated with optimisation of antimicrobial use in primary care. A systematic review and meta-analysis in 2019 [13] demonstrated that GP/CP team-based AMS implementation strategies such as group meetings, delayed prescribing, audit feedback, education, and workshops was an effective way to improve the quality of antibiotic prescribing by GPs. When the AMS roles of CPs are guided by policies that support GP/CP practice agreements and patient referrals, point-of-care testing, screening, and treatment services [14,15] and the use of patient-facing leaflets [16] in community pharmacies were feasible and substantially effective in reducing unnecessary antimicrobial use and improving patient outcomes. 

Although the literature on AMS is growing in the context of general practice and community pharmacies, AMS research on the GP–CP interface remains scant. To date, there is no AMS model framework elsewhere that guides how to better engage and promote collaboration of GPs and CPs regarding antimicrobial prescriptions and antimicrobial use. The AMS implementation frameworks found in primary care are limited to either general practice [17] or specific clinical infections (e.g., respiratory tract infections) where antimicrobials are overused for [18]. To date, there is no AMS framework that focuses on the context of community pharmacists and and GP–CP collaborative practices. In Australia, there are GP–CP collaborative care models [19,20] that support chronic care, but no such collaborative model incorporates or drives AMS activities. This study aims to develop a general practitioner/pharmacist antimicrobial stewardship (GPPAS) model for primary care in Australiathrough formative research determining effective GP–CP collaborative AMS strategies, identifying evidence practice gaps in collaborative AMS implementation practices and challenges, and revealing opportunities for GPs and CPs to collaboratively optimise antimicrobial use. 

## 2. Results

### 2.1. Key Outcomes of the Studies Contributing Evidence for the Development of the GPPAS Model

Table 1 summarises the key outcomes and key messages of the seven individual studies explored in relation to the development of the GPPAS model. Systematic reviews by Studies 1 and 2 along with updated studies in the literature identified a list of effective GPPAS interventions that could be implemented through collaboration, communication, or partnerships between GPs and CPs. The GPPAS interventions included GP-pharmacist team-based group meetings; academic detailing; the development and use of local antibiotic guidelines; antibiotic audits and feedback programmes; delayed prescribing; workshop training; use of point-of-care testing, screening, and treatment services; pharmacist-led reviews of antimicrobial prescriptions; and the use of common patient-facing antibiotic educational leaflets and checklists. Currently, the use of most GPPAS interventions has been limited in Australian primary care and there are implementation challenges associated with the GPPAS interventions (Studies 3 and 4, Table 1). Australian GPs and CPs were attitudinally optimistic about the future implementation of the emerging GPPAS interventions around AMS training, antimicrobial audits, use of AMS resources, and policies that would support GP/pharmacist practice agreements for AMS (Study 5, Table 1). The necessary system structures and systems thinking approaches for optimal implementation of GPPAS interventions were identified through Study 6 (Table 1). A pilot Study 7 reported in Table 1 found that implementation of a GPPAS submodel involving AMS education was feasible and effective for improving the appropriateness of antimicrobial selection (73.9% vs. 92.8%, RR = 1.26, 95% CI 1.18–1.34), duration (53.1% vs. 87.7%, RR = 1.65, 95% CI 1.49–1.83), and guideline compliance (42.2% vs. 58.5%, RR = 1.39, 95% CI 1.19–1.61).

### 2.2. Key Problems and Quality Improvement Strategies Informing the GPPAS Model Components

Table 2 summarises the key problems, facilitators, opportunities, and quality improvement strategies to routinely implement AMS activities through collaboration between GPs and CPs. Guided by the SEIPS 2.0 theoretical model component, we listed priority problems hindering the implementation of AMS activities, and facilitators and opportunities for addressing those problems through quality improvement strategies. The theory-guided quality improvement strategies were the building blocks for the GPPAS model framework (Table 2). Evidence-based GPPAS interventions were organised into five GPPAS implementation submodels covering education, audits, diagnostic stewardship, delayed prescribing, and routine reviews of antimicrobial prescriptions. Future improvement areas for organisational structures were identified, i.e., access to resources and policy environments that support GP-pharmacist collaboration, which would likely facilitate and foster implementation of the GPPAS submodels. The key connections among the GPPAS model components were guided by the SEIPS 2.0 theoretical model. The GPPAS model components and the generated submodels collectively contributed to the framework for the GPPAS implementation model.

### 2.3. GPPAS Implementation Model Framework: Design, Discussion, and Implications

Our study proposes a theory and evidence-informed GPPAS implementation model framework (Figure 1) which consists of seven interactive components under work systems that demonstrate how AMS programmes can be rigorously implemented in Australian primary care by fostering GP–CP collaboration. The seven interactive components include: (1) pharmacist-patient interaction, (2) GP-patient interactive communication, (3) GP-pharmacist collaboration, (4) resource access (tools and technology), (5) organisational structure, (6) tasks, and (7) a policy environment that fosters interpersonal and interprofessional GP–CP collaboration in AMS. The GP/pharmacist domain of the GPPAS model is composed of the following five GP/pharmacist team-based submodels for routine AMS implementation: interprofessional AMS education, antimicrobial audits, diagnostic stewardship, delayed prescribing through GP-pharmacist partnerships, and routine antimicrobial prescription reviews. Implementation of these submodels would foster collective participation and collaboration of GPs and CPs in AMS. Overall, improving the AMS work system would change cognitive and social behaviours at the patient, professional, and practice levels to achieve the desired outcomes in primary care, i.e., optimal antimicrobial use, reduced patient harm by avoiding unnecessary antibiotic use, and reduced risk of increased antimicrobial resistance.

#### 2.3.1. GPPAS Component 1: Pharmacist-Patient Interactions

A community pharmacy is the first point of contact for most patients who seek care or advice from a pharmacist for treating minor ailments or self-limiting infection(s). Furthermore, CPs are the most accessible sources for patients to get antimicrobial(s). Pharmacist-patient interactive communications, thus, can play an important role in educating patients about how to prevent minor infections without antimicrobial(s) and safe use of antimicrobial(s). As guided by the SEIPS 2.0 model, an interpersonal trust and dependency between a CP and a patient about antimicrobial-use decisions is critical to reduce a patient’s behaviours of desiring or demanding antimicrobial(s). Pharmacist-patient interactions can impact individual knowledge, provide motivation to act, and change the social norms of both the CPs and patients regarding safe antimicrobial use in primary care.

In conjunction with the international evidence [27,28,29,30,31,32,33,34], our GPPAS model emphasises the routine provision of essential AMS roles of CPs when patients visit a pharmacy. The roles include (i) diagnostic assessment of infections to an extent to appropriately identify a patient’s needs (e.g., OTC, antimicrobial, and referral to GPs); (ii) patient education, where possible, about self-limiting infections, completion of the full course of antimicrobial therapy as prescribed, avoiding the use of leftover antimicrobial(s), and the effects of antimicrobial use on the gut; (iii) asking a patient to call a pharmacist if any allergies or side effects occur during the period of antimicrobial use; (iv) providing patients with instructions about returning leftover antibiotics to pharmacists.

Appendix A shows evidence [16,35,36,37] of the effectiveness of resources (such as antibiotic checklists) used by CPs to support their AMS roles. These tools can influence appropriate patient referrals to GPs, and antibiotic-seeking behaviours and antibiotic consumption by patients. Most CPs in Australia provide patient education about antimicrobial(s) and resistance; however, they do not use any formal communication tools as these are not readily available in practice sites, as reflected in our study [31]. The availability of empirical diagnostic resources and patient educational resources in community pharmacies could foster CP-patient relationships and trust for joint antimicrobial-use decisions, spreading antibiotic awareness at the community level for marked impacts.

#### 2.3.2. GPPAS Component 2: GP-Patient Interactive Communication

Our GPPAS model stresses that GP-patient interactive communication is as important as CP-patient interaction to collectively identify patient concerns and reduce patient pressure on antibiotic decisions for infections which can be treated without antibiotics or antimicrobials. There is an importance of GP-patient shared decision making on antibiotics. A GP can use CPs an an extended arm, and refer patients to CPs for complete understanding of the antibiotic courses, antibiotic use awareness, food and drug interactions with antibiotics, and possible side effects. This multidisciplinary interactive effort would reduce patient pressure and improve community awareness about optimal antibiotic use. The national and international evidence [17,38,39,40,41] supporting GP/patient interactions is reported in Appendix A.

#### 2.3.3. GPPAS Component 3: GP-Pharmacist Collaboration

An interdisciplinary approach [42,43,44] is key to effectively implementing AMS programmes. Our study evidence proposes five intervention-focused GPPAS implementation submodels under the GP/pharmacist collaboration component of the GPPAS model framework. The GPPAS implementation submodels were proposed for effective implementation of (i) antimicrobial audits and feedback programmes, (ii) AMS educational programmes, (iii) delayed prescribing programmes, (iv) routine review of antimicrobial prescriptions programmes, and (v) diagnostic stewardship programmes.

##### GPPAS Implementation Submodel for Antimicrobial Audits and Feedback Programmes

An antimicrobial audit and feedback programme is a proven, effective, and sustainable model [13,45] where GPs and CPs need greater involvement to improve the quality of antimicrobial use. As our studies [22,23] found, GPs and CPs both demonstrated positive attitudes towards interprofessional involvement in antimicrobial audits. Although auditing antimicrobial prescriptions is common in Australian hospital settings, resources and the structural set-up for regular audits in primary care are potentially limited and underdeveloped, which include: quality indicators for the optimal use of antimicrobials, electronic databases that facilitate clinical and antimicrobial audits, a dedicated interprofessional team, and context-specific tools supporting audits of antimicrobial prescriptions. Although ambitious, development in the area could help primary care pharmacists including clinical pharmacists who are co-located in Australian general practices to be more engaged in regular antimicrobial audits, reviewing stewardship indicators and providing related communication and feedback to GPs. Considering shortages and role fitness of health professionals in primary care, upskilling CPs about antimicrobial audits would facilitate normalization of this GPPAS strategy at local level by creating local GP–CP network. In Australia, an antimicrobial audit tool [46] has been tested in the remote primary healthcare which could be integrated into GP practices in the future. Considering the feasibility of the model, audits and written feedback to GPs could be undertaken for selective antimicrobial prescription(s) such as prescriptions for RTIs or UTIs (where antibiotic misprescribing is higher) but with a regular interval (e.g., monthly or quarterly a year). Evidence [47,48] of such pharmacist-led effective audit models in primary care is summarized in Appendix A.

This model suggests that a pharmacist would collaborate with the local AMS team to identify the educational needs, to build programme initiatives, to plan for data collection, and to share and discuss the audit-feedback programme outcomes. Sustained funding and involvement of an infectious disease physician, AMS pharmacist, and GP opinion leader to support audit programmes should be a policy decision. In Australia, a few GP clinics, where pharmacists are co-located, could be settings for testing this model as a priori. The implementation feasibility of this model would require better access by general practice and community pharmacist to an AMS pharmacist or infectious disease physician where possible and relevant education, training and practice protocol for CPs.

##### GPPAS Implementation Submodel for AMS Educational Programmes

AMS educational and training programmes for GPs and CPs were found to be essential both in the literature [11,17,18,49,50,51,52] and in our baseline evidence [22,23] with respect to undertaking AMS programmes in routine patient care. Although AMS training is a core component of AMS programmes, training that targets primary care clinicians is limited; only one-third of our surveyed GPs and CPs in Australia had completed the antimicrobial modules of the NPS Medicine Wise [22,23]. This limited uptake demonstrates the implementation challenges of AMS training among GPs and CPs. Van Katwyk et al. [52] identified 94 AMS-related educational programmes globally, however, few of them were accredited training programmes. The feasible implementation and sustained impact would have been desirable if these training courses had been incorporated and regulated into the graduate curriculum of GPs and CPs [53].

A platform that may help GPs and CPs to learn about AMS from each other’s professional perspectives is lacking in Australia. We propose a GPPAS educational model that can be defined as a community-based model where GPs and CPs would co-construct AMS knowledge by sharing areas of problematic antimicrobial prescribing decisions, exploring dissonance in opinion, and developing consensus about a safer antimicrobial therapy between themselves. The system development approach and theories of interprofessional education back up this model. The support from AMS physicians and AMS pharmacist and their networking are important to newly establish this model in any setting.

The proposed model would likely develop a sustainable interprofessional AMS learning process and platform, by utilising the expertise of pharmacists on antimicrobial pharmacotherapy, pharmacokinetics, pharmacodynamics, dosing, antimicrobial spectrum, and resistance to influence GPs’ antimicrobial prescribing. International evidence of such a model [53,54,55,56] is reported in Appendix A. In Australia, a GPPAS educational model was shown to be significantly effective in improving appropriateness in antimicrobial selection (73.9% vs. 92.8%, RR = 1.26, 95% CI 1.18–1.34), duration (53.1% vs. 87.7%, RR = 1.65, 95% CI 1.49–1.83), and guideline compliance (42.2% vs. 58.5%, RR = 1.39, 95% CI 1.19–1.61) [26].

The mode of implementation involves GP/pharmacist regular group meetings and GP/pharmacist co-led workshop training. Research has shown that GPs and CPs in Australia are motivated to be involved in the co-construction of AMS knowledge by collaborative learning processes [24]. A GP–CP collaborative group meetings model was interprofessionally supported (GPs vs. CPs; 54.9% vs. 82.5%) to optimise antimicrobial therapy, although there was an attitudinal divergence [24]. This result indicates that GPs should be open to interprofessionally sharing and accepting interventions when justified, in order to improve the quality of antimicrobial prescription(s) and to learn as a team. Creating a culture that supports co-constructing AMS knowledge is important in order to develop a sustainable interprofessional AMS learning platform in primary care.

##### GPPAS Implementation Submodel for Delayed Prescribing Programmes

The antimicrobial delayed prescribing strategy has been proven to be an effective AMS approach [57,58] to impact antimicrobial dispensing rates [59]; however, its true effect on reducing antimicrobial use by patients remains unclear. This is perhaps due to the fact that GPs may delay prescribing antimicrobial(s) but CPs might not delay dispensing antimicrobial(s) for reasons such as a patient’s demand for early dispensing or a CP’s strong commitment. This raises questions about the implementation process of this strategy. According to our GP-survey study [22], most GPs (72.2% of 385 GPs) frequently used this strategy. Another study [60] with 103 Queenslander CPs in Australia reported that 40% would dispense a delayed antibiotic prescription within 24 h of a GP visit by a patient and 60% of CPs would not. CPs could play an important role in implementing delayed prescribing strategy, but they are still under-used resources in Australia. We propose a local GP–CP partnership model to foster implementation of a delayed prescribing strategy and to harness the involvement of CPs in the strategic implementation in primary care. While GPs prescribe delayed antimicrobial, a highlighted instruction somewhere on the prescription to direct CPs on how many days should be delayed before dispensing this prescription should be included. CPs would communicate with GPs to reassure if a patient demands early dispensing. This initiative may assist effective implementation of this strategy through some collective effort from GPs and CPs.

##### GPPAS Implementation Submodel for Review of Antimicrobial Prescriptions

The routine review of antimicrobial prescriptions is a useful patient-centric approach to optimise antimicrobial therapy. Thus, routine reviews have become an important part of AMS activities by pharmacists because they identify many aspects of prescribing as compared with a passive antimicrobial audit model. A model in which CPs review GPs’ antimicrobial prescriptions is likely to provide active learning and teaching opportunities for CPs and GPs to adopt AMS activities in routine practices. A key advantage of this model is that each antimicrobial prescription can be assessed for patient safety, AMR risk, and cost savings.

General medication reviews are conducted in community pharmacies; however, related training and a checklist that would guide CPs about when, what, and how to commence reviewing antimicrobial prescriptions are not readily available. This might be one of the reasons why CPs less frequently assess guideline adherence of antimicrobial prescriptions and are less likely to communicate with GPs when the choice of an antimicrobial is believed to be suboptimal. Our study found that more than 40% of CPs did not feel confident to assess guideline adherence of prescribed antimicrobial(s). This post-prescription review role of CPs should be a part of routine pharmacy practice to avoid antimicrobial medication errors and to ensure guideline compliance. Advocating these essential roles of CPs could create an opportunity to produce AMS pharmacist stewards, who could support CPs to develop AMS skills and eventually sustainably reduce some problematic prescribing. Using a GP/CP team-based routine antimicrobial prescription review model is one of the approaches that needs future development and evaluation.

Pragmatically, an antimicrobial review can be specific to an infection, a patient category, or the type of antimicrobial prescribed (e.g., broad-spectrum). Access to a patient’s clinical indication and diagnostic reports including antibiograms would be important to facilitate this model. For CPs who work as home medicine review pharmacists and co-located pharmacists in general practice, testing this model in those settings might be an immediate practical step forward. The role of CPs in antimicrobial reviews has been supported by the international literature [11,32,33,34,61] (Appendix A).

##### GPPAS Implementation Submodel for Diagnostic Antimicrobial Stewardship

Diagnostic uncertainty is one of the most cited reasons for inappropriate use of antimicrobials in a community [62,63,64]. Practically, it is often a challenge for GPs and CPs to differentiate between bacterial and viral infection(s) based on apparent clinical symptoms. Diagnostic uncertainty causes GPs to unnecessarily prescribe an antimicrobial for infections that are treatable without antimicrobials, and GPs err on the side of caution [65]. Similarly, due to diagnostic uncertainty, CPs inappropriately and unnecessarily refer patients to GPs, increasing the risk of patients receiving an antimicrobial prescription and/or the likelihood of dispensing antimicrobial(s) when it is not warranted. We propose a GP–CP collaborative diagnostic stewardship model in Australia to minimise unnecessary use of antimicrobials caused by diagnostic uncertainty. Appendix A describes evidence and use of these tests in the international context [7,14,66,67,68,69,70,71].

The uptake of point-of-care testing by the Australian GPs and CPs was below 20% according to our 2019 nationwide surveys. The evidence indicates that an improved patient referral system, collaborative practice agreements, and user training would help to introduce a new GP–CP collaborative diagnostic stewardship model in Australia. This model would have the following implications: (i) increasing two-way appropriate patient referrals between GPs and CPs and (ii) ensuring that a patient needs either antibiotics or over-the-counter (OTC) or GPs/CPs visits to optimally treat infections. The provision of this model could improve the scope of CPs to be better involved in AMS programmes and to optimise infectious disease patient care through community pharmacies.

#### 2.3.4. GPPAS Component 4: Tools and Technology (Access to Resources)

The active involvement of GPs and CPs in AMS programmes and the implementation of GPPAS strategies require: relevant resources, tools, and technologies; a clinical decision support system prescribing and dispensing software integrated with updated antibiotic guidelines, regional antibiograms, and AMR reports; “My Health Records” providing sufficient clinical and medication information for review by pharmacists; point-of-care tests facilities; the WHO AWaRe antibiotic tool; patient-facing information leaflets; and a checklist defining GP/pharmacist interprofessional AMS activities. These resources will increase the awareness, motivation, and confidence of GPs and CPs to effectively conduct routine AMS tasks. For instance, the use of patient-facing leaflets is an opportunity for both GPs and CPs to provide safety net advice to patients, to revise the decision of antimicrobial use, and to address demands for antimicrobials by patients. Addressing the gaps in tools and technologies would have a significant influence by shaping the organisational set-up for GP–CP collaboration regarding the AMS and implementation of the GPPAS submodels.

#### 2.3.5. GPPAS Component 5: Organisational Support (System Structures)

Organisational support has been deemed as important to set up a GPPAS model for routine patient care and to support GPs and CPs to collaborate with antimicrobial prescription(s). The GP/pharmacy organisational initiatives which have been found as priorities to facilitate implementation of evidence-based GPPAS strategies include: (i) the development of antimicrobial use and resistance data sharing platform between GPs and CPs, (ii) building a local GP–CP network for collaboration, (iii) pharmacist’s access to a patient’s health records including antimicrobial prescriptions, (iv) general practice/community pharmacy practice agreements for AMS, (v) interprofessional organisational structures that ease patient referrals and enable GPs and CPs to follow-up on patients who are using antimicrobials, and (vi) organisational supports for GP–CP collaboration in AMS. There are structural limitations to routinely monitoring the appropriateness of prescribed antimicrobial(s), patients’ clinical and therapeutic outcomes, and providing feedback to GPs. Implementation of a GP/pharmacist antimicrobial audit and feedback model and a routine antimicrobial prescription review model, thus, need a sustainable data solution. The SEIPS 2.0 model component, “tools and technology” recommends that these data should fit into the workflow of GPs and CPs.

#### 2.3.6. GPPAS Component 6: Tasks

The potential routine collaborative tasks of GPs and CPs for implementing AMS include: the use of antibiotic guidelines and antibiotic checklists, recording clinical and therapeutic details of patients in available electronic health records and “My Health Records”; ensuring that delayed prescribed antimicrobials are delayed dispensed; GP/pharmacist communication with antimicrobial prescription whenever required to address inappropriate use; use of point-of-care diagnostic stewardship tools (e.g., c-reactive protein (CRP) or rapid antigen testing (RADT) for Group A streptococcus (GAS) pharyngitis) to avoid unnecessary use of antibiotics and appropriate patient referral; review of each antimicrobial prescription by CPs and communication with GPs to address any inconsistency in choice, dose, and dose regimen; attempting shared decision making with patients in situations of antimicrobial use to treat infection(s); and providing patient education during consultation using decision support tools such as antibiotic checklist. Strengthening interconnected domains of the GPPAS model would facilitate these routine tasks according to the SEIPS 2.0 model.

#### 2.3.7. GPPAS Component 7: Policy Environment

Having a national governance structure for implementing AMS programmes in primary care is a policy priority. GP and pharmacy representatives in the governance structure would determine and provide a strategic direction to foster GP–CP collaboration in AMS. The external environment such as GP/pharmacy AMS practice guidelines and protocols may drive implementation. The adoption of societal norms and policy on restricted use of broad-spectrum antimicrobials, stopping repeat prescriptions, establishing GP/pharmacy practice agreements, and incentives and/or medicare support for local GP–CP collaboration in relation to antimicrobial prescription and infectious disease management programmes could influence AMS progress in primary care. These societal interventions and policies need greater attention by AMS stakeholders to support implementation of GPPAS model in primary care.

#### 2.3.8. GPPAS Implementation Process and Outcomes

According to the SEIPS 2.0 theoretical model, if an AMS supportive work system (Figure 1) has been set up in GP/pharmacy practices, the professional AMS activities and collaboration will be accelerated through cognitive, social, and behavioural changes among GPs and CPs. This process would influence the achievement of the desired outcomes: improving uptake of evidence-based AMS strategies by GP and CPs, optimising the use of antimicrobials, improving patient outcomes, and reducing patient harm and resistance.

## 3. Discussion

The implementation of AMS programmes in primary care is potentially limited, although most antimicrobials are used in primary care. Acknowledging that primary care practices are everchanging, diverse, and influenced by a lack of interprofessional health service models, it is imperative to develop a sustainable AMS implementation model involving GP–CP collaboration in primary care. Our study, for the first time, has designed a framework for an evidence-based novel GPPAS implementation model that provides important insights into how Australian GPs and CPs could be better engaged in AMS to fight against growing AMR in primary care. The framework for the GPPAS model includes five implementation submodels to foster AMS programmes through improved collaboration between GPs and pharmacists in primary care.

The GP/pharmacist interprofessional AMS educational submodel is emerging. Such a model in Scotland was effective for reducing the use of broad-spectrum antimicrobials in a large region as part of a national initiative [54]. A system-supported GP–CP collaborative pharmacotherapy audit meetings model demonstrated improvemed antibiotic prescribing in a randomised controlled trial [55]. A multimodal AMS educational programme involving GPs and pharmacists showed a long-term impact on sustained reduction in antibiotic prescribing and infections caused by *E. coli* in a community [56]. In the UK, a GP/pharmacist consensus-based national AMS competencies curriculum has been developed for undergraduate GP/pharmacy professionals [53]. Incorporation of shared case-based AMS learning modules into respective GP/pharmacy curricula may help educate future generation GPs and CPs about the importance of interprofessional engagement in AMS. In Australia, a GPPAS AMS educational submodel was well received, cost effective, and showed potential in a pilot study for improving appropriateness and guideline compliance of antimicrobial prescribing by GPs [26]. Whether and how long these improvements are sustained after ceasing GPPAS intervention involving AMS education is also worth exploring; such a study has been conducted and will be shortly published by our research team elsewhere.

To facilitate similar interprofessional AMS educational programmes in Australia, development of a local GP/CP interprofessional team as site champions is a necesary step for implementation. Incorporating AMS educational submodels into existing GP/pharmacist collaboration models such as pharmacist co-located in general practice model [19] and a home medicine review pharmacist model [20] could be worth trying where a growing number of CPs work in collaboration with GPs.

Our GPPAS model framework proposes a GPPAS implementation submodel for antimicrobial audits and feedback which has enormous importance for antimicrobial surveillance, monitoring and identifying evidence practice gaps and developments of stewardship targets in primary care. Such a model [47] led by an ambulatory care pharmacist demonstrated effectiveness in improving antibiotic prescribing for upper respiratory tract infections and urinary tract infections in the United Sates [47]. Another study [48] showed that this model, when led by an AMS physician and an AMS pharmacist together, was highly effective to significantly improve guideline-concordant antibiotic prescribing from 38.9% to 57.9% in a family medicine clinic. Authors [48] also reported significant improvements in the selection (68.9% to 80.2%), dose (76.7% to 86.2%), and duration of antibiotic therapy (73.3% to 86.2%) according to antibiotic guideline. We identified gaps in the comprehensive and quality data for conducting local and national level antimicrobial audits. We emphasise the importance of developing data infrastructures and building interprofessional teams involving primary care pharmacists to run this GPPAS sub-model at the local and national levels.

Diagnostic uncertainty still affects Australian GPs and CPs to avoid unnecessary use of antibiotics for treating infections such as RTIs and UTIs. Our GPPAS diagnostic stewardship submodel would help address diagnostic uncertainty of GPs and CPs potentially for patients with RTIs and UTIs. The use of point-of-care testing (.e.g., C-reactive protein) for and rapid antigen diagnostic tests) for RTIs including acute sore throat caused by Group A streptococcal infections has been growing for appropriate clinical decisions on antimicrobial therapy in primary care [66,67]. Having these diagnostic tools both in general practice and in community pharmacies can be a source of collaboration between GPs and CPs. These tools would support appropriate patient referrals and selection of appropriate antimicrobials when delayed antimicrobial prescription might have undesired consequences for patient outcomes [68]. The use of point-of-care testing, screening, and treatment services in U.S. community pharmacies utilising local GP/pharmacy practice agreements has demonstrated effectiveness in reducing antimicrobial use in pharyngitis, bronchitis, and influenza management [7,14,69,70,71]. The model was feasible and acceptable to patients without any patient harm. The state-wide use of these services is increasing in pharmacies in the United Kingdom and the United States [7,14,69,70,71] as compared with in Australia. The contextual feasibility and cost-effectiveness evidence are much needed for policy actions in Australia to introduce our GPPAS diagnostic stewardship model in primary care. The policy initiatives for GP/pharmacy practice agreements could help implement proposed stewardship models at the local and state levels.

Our GP–CP partnership based submodels accelerating implementation of delayed prescribing and routine review of antimicrobial prescriptions by pharmacists could have marked impacts. Australian GPs and CPs were enthusiastic about a partnership-based delayed antibiotic use strategy, and the majority of GPs were receptive to receive CPs’ recommendations on the choice, dose, and duration of antimicrobial therapy where appropriate. Improvements in the organisational structures and technologies that support GP–CP collaboration, and advancement of “My Health Records” could facilitate routine implementation of those submodels.

The routine use of antibiotic checklists and leaflets supporting CP-patient and GP-patient interactions for shared decision making on antimicrobial use seems promising. A randomised controlled trial study demonstrated that the use of patient-facing resources (e.g., take-home leaflets for patients) during CPs’ routine consultations was associated with increased self-care advice and decreased patient referrals to GPs [16]. The provision of advising patients that “antimicrobials do not reduce the severity and duration of symptoms” and “the symptomatic relief is the best option to help recover the condition” might bring benefits in improving patients’ antibiotic-seeking behaviours for treating minor infections [35]. In an Australian study, parents stated that they would visit multiple GPs if they believed that an antibiotic was required for their child [39]. This indicates that antibiotic prescriptions can be avoided in many cases if GPs can identify the concern and real expectations (e.g., symptom relief and worry about the severity of disease) of the clients or patients. The resources that outline how to empower patients in therapeutic decisions for treating infection and communication skill training would be beneficial [17,40,41]. The CP-patient and GP-patient interactive components of the GPPAS model framework highlights the importance of using interactive tools to improve shared decisions on antimicrobial therapy.

In Australia, apart from developing AMS resources and technologies, a legal framework [72] can be instrumental for institutional support from GP/pharmacy practices to organise, manage, and implement GPPAS implementation submodels involving antimicrobial audits and feedback programmes, point-of-care testing, and review of antimicrobial prescriptions by CPs. The Core Elements of Outpatient Stewardship developed by the Centre for Disease Control in USA highlighted the involvement of pharmacists, both with and without infectious disease training, as key team members to facilitate evidence-based AMS strategies including antimicrobial audits [73]. The Australian national regulatory framework for RACGP accreditation [74,75] and the national competency standards framework for pharmacists in Australia [76] should incorporate AMS programmes as an accreditation criterion to foster implementation of AMS through GP-pharmacist collaboration in primary care. The professional GP (e.g., the Royal Australian College of General Practitioners (RACGP)) and pharmacy (e.g., Pharmaceutical Society of Australia (PSA)) organisations, NPS Medicine Wise, and the Primary Health Network (PHN) have major roles to collectively work to develop governance structure and to seek policy support from the government to foster implementation of GPPAS submodels through liaisoning with AMS stakeholders and policymakers.

This study has some drawbacks. Methodologically, an in-depth qualitative study (e.g., paired interviews) and multi-centered pilot study would also have been ideal to refine the model components and to inform future randomised controlled trials. However, those studies were out of the scope of this project. Although the SEIPS 2.0 model established theories for the GPPAS model, other frameworks such as the theoretical domain framework [77] and implementation science theory [78] might have been used as alternative approaches to make this model work in real life because of the complex nature of AMS interventions and diverse primary care practices. The detailed limitations of the seven formative studies that contributed evidence for the GPPAS implementation model have been explained in our previously published literature [13,21,22,23,24,25,26].

Future research is needed to refine the GPPAS model framework using a co-design approach.The implementation evaluation of the GPPAS model, and the submodels needs clinical trials and qualitative studies supported by grounded theory or behavioural approaches [79,80,81]. Lastly, our GPPAS implementation model could be utilised as a guide for stakeholders and policymakers to design and implement AMS programmes utilising GP–CP collaboration in primary care both nationally and internationally. The model could have significant policy implications to define, involve, and foster the much-required role of CPs in the implementation of AMS programmes globally.

## 4. Conclusions

A GPPAS implementation model framework has been successfully designed which will guide and accelerate the implementation of evidence-based AMS programmes by improved GP–CP collaboration in Australian primary care. The GPPAS model could have significant policy implications to foster GP–CP collaboration in AMS. Future implementation trials are needed to deeply understand the feasibility and cost effectiveness of the GPPAS model for national scalability to sustainably optimise antimicrobial use in primary care.

## 5. Methods

An exploratory study design was used to obtain evidence for a prototype GPPAS implementation model. The exploratory studies had different aims and used respective methodologies of systematic review, scoping review, rapid review surveys, qualitative study, and a pilot study with a before-after design. The studies were conducted between 2017 and 2021; the secondary evidence was collected through reviews and the primary evidence in the Australian context was collected through nationwide surveys and a pilot study. Saha et al. (2021) [82] detailed the methodologies that guided the development of the GPPAS model. Mutiple reviews [13,21,25] were undertaken to identify and describe the list of effective GPPAS interventions and model components. Our systematic review [13] used the Template for Intervention Development and Reporting (TIDIeR) checklist to describe GPPAS interventions. AMS surveys of GPs and CPs across Australia were designed to explore the convergent and divergent views of GPs and CPs about GPPAS interventions, attitudes towards collaboration in AMS, and the perceived challenges of implementing GPPAS interventions. A pilot study in an Australian general practice [26] determined the effectiveness of a GPPAS submodel. Summary evidence was synthesised from all studies and a framework analysis was conducted using the theoretical framework described below to inform the prototype GPPAS model framework. Figure 2 diagrammatically shows how data from the seven studies contributed to the development of the framework for the GPPAS model.

### 5.1. Theoretical Model Selection

Theory- and evidence-based approaches were employed to develop a GPPAS model. We identified the Systems Engineering Initiative in Patient Safety (SEIPS) 2.0 model [83] as a vehicle for translating our results into a model that would guide how to foster AMS implementation in primary care through GP–CP collaboration. The SEIPS 2.0 model guided the theoretical structure of the GPPAS model by identifying the determinants of GPPAS uptake, as well as barriers and facilitators to implementing AMS through GP–CP collaboration.

Human factors engineering is the scientific approach to understand interactions among humans and the elemental parts of a system [84]. The SEIPS 2.0 model is one of the leading frameworks in the discipline of human factors engineering research [84,85]. According to the SEIPS 2.0 model, the workplace (e.g., GP clinic or community pharmacy) characteristics and a work system interact spontaneously. This model identifies the problems in a work system that consist of person(s), tools, technologies, organisation, tasks, and the physical environment within a wider societal environment. The work system influences the processes or practices by health professionals (e.g., GPs or pharmacists) such as prescribing and dispensing practices of antimicrobials that influence the outcomes of patients with infection(s) and impact AMR.

The SEIPS 2.0 model has been increasingly applied to improve the safety of patients and the quality of care in various healthcare settings [86,87,88]. More than fifty studies have used the model for assessing the safety of patients in healthcare, ranging from hospital to primary care clinics. Importantly, the SEIPS 2.0 model can be used to evaluate the causes of medical errors and the process of how to control them [87]. Hence, this model is a natural fit for studying the behavioural and systematic components of AMS involving GP–CP collaboration for safe antimicrobial use. This model helps to identify interventions at an individual, structural, and an environmental level to foster AMS through GP–CP collaboration. The SEIPS 2.0 model, therefore, was used to determine the building blocks of the GPPAS model framework.

### 5.2. Evidence Synthesis and Analysis

A critical synthesis approach was used to collect evidence from multi-method research [88]. We merged and triangulated evidence derived from our published studies (Table 1) using a SEIPS 2.0 theoretical framework. A team of researchers responsible for each individual study summarised the key outcomes for each study: effective GPPAS strategies, commonly and uncommonly used GPPAS strategies, barriers and facilitators to collaboratively implement GPPAS strategies by GPs and CPs, and attitudes towards GP–CP collaboration in AMS. Where possible, the authors evaluated and compared evidence from multiple sources to weigh the evidence. Evidence from systematic reviews and meta-analyses and evidence from multiple sources were considered to be strong evidence. In Section 2, the designed GPPAS model was described with supportive evidence from multiple sources. The evidence synthesised from a pilot study of a GPPAS model component provided the contextual evidence in Australian primary care. Upon triangulation, we identified key model components and drew the structure of the GPPAS model framework. Then, the intervention-based GP–CP collaborative implementation submodels were mapped out under the GPPAS model framework. The required resources and policies were identified and proposed to integrate the submodels into anexisting GP/pharmacist interprofessional care model in Australia. The amalgamation of synthesised evidence eventually informed a prototype GPPAS implementation model framework. The Guidance for Reporting Intervention Development Studies in Health Research (GUIDED) checklist [89] was used for quality and consistency of GPPAS intervention development reporting.

## Figures and Tables

**Figure 1 antibiotics-11-01158-f001:**
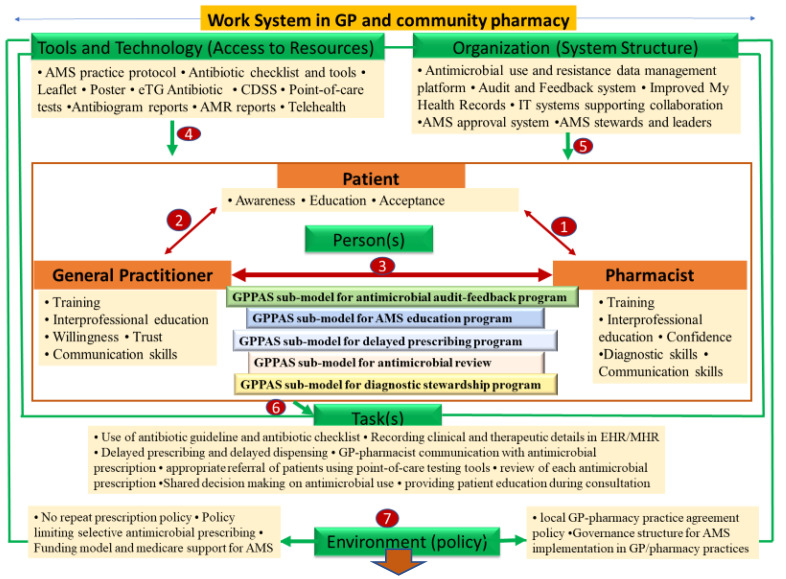
GPPAS implementation model framework guiding AMS implementation by fostered GP-pharmacist collaboration. [eTG: electronic Therapeutic Guideline; CDSS: Clinical Decision Support System; IT: Information technology; MHR: My Health Record; EHR: Electronic Health Record; AMR: Antimicrobial Resistance; AMS: Antimicrobial Stewardship; GPPAS: General Practitioner-Pharmacist Antimicrobial Stewardship].

**Figure 2 antibiotics-11-01158-f002:**
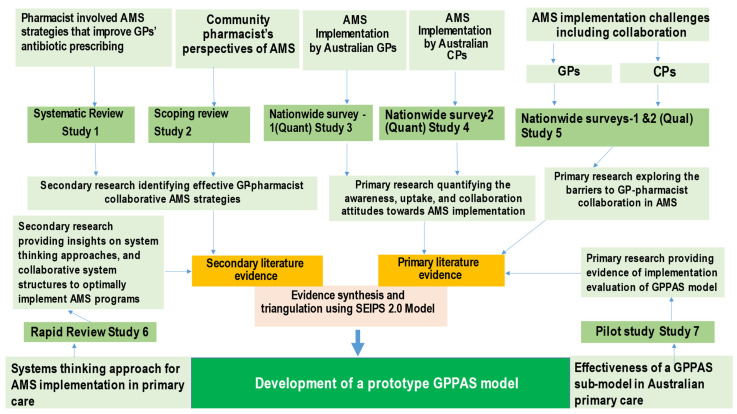
Stepwise development of the GPPAS model. Secondary research indicates reviews collecting secondary data from published studies. Primary research indicates studies that collected primary data in the context of Australian primary care.

**Table 1 antibiotics-11-01158-t001:** Summary of key findings from seven individual studies that contributed evidence for the development of the GPPAS model.

Studies That Contributed Evidence for the GPPAS Model	Methods	No. of Studies/Participants (Settings)	Aim	Key Outcomes	Key Message
**Study 1:** Effectiveness of Interventions Involving Pharmacists on Antibiotic Prescribing by General Practitioners: A Systematic Review and Meta-Analysis [13]	Systematic review and meta-analysis	35 Studies General practice	To identify which interventions involving pharmacists could improve antibiotic prescribing by GPs	A meta-analysis of 15 studies found a reduction in the antibiotic prescribing rate (odds ratio 0.86 and 95% CI 0.78–0.95) and an improved guidline-adherent antibiotic prescribing rate (odds ratio 1.96 and 95% CI 1.56–2.45) when interventions were implemented by a GP/pharmacist team. A list of effective GPPAS strategies included: (1) GP/pharmacist group meetings, (2) academic detailing by a GP/pharmacist team, (3) the development of GP–CP collaboration development and use of local antibiotic guidelines, (4) auditing antibiotic prescriptions and providing feedback to GPs, (5) implementing a delayed prescribing strategy by a GP/pharmacist partnership, (6) workshop training involving GPs and pharmacists, (7) use of point-of-care tests in a GP/pharmacist practice agreement model, (8) reviewing of antimicrobial prescriptions by pharmacists and communication with GPs, and (9) use of common patient educational leaflets and antibiotic checklists.	GPPAS interventions were effective to reduce antibiotic prescribing and improve guideline-adherent antibiotic prescribing by GPs. However, context specific evidence is required to understand their usage in routine practice and implementation barriers.
**Study 2:** Knowledge, Perceptions, and Practices of Community Pharmacists towards Antimicrobial Stewardship: A Systematic Scoping Review of Survey Studies [21]	Scoping review	10 Studies Community pharmacy	To review AMS survey studies and to determine the knowledge, attitudes, and practices of community pharmacists regarding AMS	This review identified a small (10 surveys globally) but growing body of survey studies in the literature that concerned CP-AMS. The dearth of surveys (only three were validated surveys) indicated suboptimal progress of AMS implementation in community pharmacies and a need for future studies to comprehensively understand the AMS situation in Australian community pharmacies. CPs believed that AMS improved patient care and reduced inappropriate antibiotic use. CPs had limited communications with prescribers about infection control and the uncertainty of antibiotic treatment. Nearly half of the surveyed CPs educated patients. The most commonly reported barriers to AMS implementation involved a lack of training, practice guidelines, access to prescribers, and reimbursement models.	There is a limited number of good quality validated AMS survey instruments around the world to assess CPs’ knowledge, use of evidence-based AMS strategies, collaboration with prescribers to identify stewardship targets, and to monitor stewardship progress in community pharmacies. The mechanism on how to improve engagement of CPs in AMS needs more research in the future.
**Study 3:** A Nationwide Survey of Australian General Practitioners on Antimicrobial Stewardship: Awareness, Uptake, Collaboration with Pharmacists and Improvement Strategies [22]	Nationwide survey	386 GPs General practice	To assess GPs’ awareness of AMS, uptake of AMS strategies, attitudes towards GP-pharmacist collaboration in AMS, and perceived challenges of doing AMS activities in routine practices	Most GPs were familiar with AMS. Two strategies were found to had increased uptake: the use of therapeutic guidelines “Antibiotics” (83%) and delayed prescribing (72.2%). Point-of-care testing (18.4%), patient information leaflets (20.2%), peer-prescribing reports (15.5%), and audits with feedback programmes (9.8%) were rarely used. Among the GPs, 50% were receptive to recommendations by pharmacists on the choice of antimicrobial and 63% were receptive to recommendatins by pharmacists on the dose. A policy fostering increased GP-pharmacist collaboration in AMS was supported by more than 60% of the surveyed GPs. Patients’ quick recovery desires broad recommendations in the antibiotic guideline, limited access to ID physicians andpharmacists, and prompt microbiological services, decision support tools, and the lack of education and training on AMS programmes were common reported barriers to optimal antibiotic prescribing.	GPs were aware of AMS but the implementation of evidence-based AMS strategies was inadequate. The majority of GPs were receptive to a pharmacist’s interventions to optimise antimicrobial use. The development of a feasible GP/pharmacist collaborative AMS implementation model, supplying stewardship resources and facilitating training could improve GPs’ participation to foster AMS activities in primary care.
**Study 4:** Antimicrobial Stewardship by Australian Community Pharmacists: Uptake, Collaboration, Challenges, and Needs [23]	Nationwide survey	613 Community pharmacists (CPs) Community Pharmacy	To assess CPs’ awareness, uptake of evidence-based AMS strategies, attitudes toward collaboration with GPs in AMS, and barriers to improving AMS practices in pharmacies	Although CPs were familiar, they felt there was a need for training (76.5%) and access to AMS practice guidelines (93.6%). CPs often counseled patients and reviewed drug interactions (93.8%) but less frequently used the national Therapeutic Guideline (Antibiotic) (45.5%) and assessed the guideline compliance of prescribed antimicrobials (37.9%). CPs inadequately communicated with GPs (41.8%) regarding suboptimal antimicrobial prescription. CPs believed that GPs were non-receptive to their recommendations. CPs strongly believed that GPs should accept their recommendations on choice (82.6%) and dosage (68.6%). CPs uncommonly used the point-of-care tests (19.1%) and patient information leaflets (24.5%). Most surveyed CPs strongly supported policies regarding GP-pharmacist collaboration (92.4%), limiting accessibility of specific antimicrobials (74.4%), and reducing repeat dispensing of antimicrobial (74.2%). CPs identified interpersonal, interactional, structural, and resource-level barriers to spontaneously participate in AMS activities.	CPs are aware of the judicious use of antimicrobials but they need training and resources to routinely practice AMS. The receptiveness of GPs and a GP–CP collaboration system structure may accelerate CPs’ engagement in AMS.
**Study 5:** Divergent and Convergent Attitudes and Views of General Practitioners and Community Pharmacists to Collaboratively Implement Antimicrobial Stewardship Programmes in Australia: A Nationwide Study [24]	Nationwide survey	999 Participants Quantitative responses: 386 GPs and 613 CPs Qualitative responses: 221 GPs and 592 CPs General practice and community pharmacy	To unveil GPs’ and CPs’ convergent and divergent attitudes regarding implementation of AMS and to understand challenges of collaboration in AMS issues between GPs and CPs	The need for AMS training by CPs was significantly higher than GPs (p < 0.0001). GPs used Therapeutic Guideline (Antibiotic) at much higher rate than CPs (p < 0.0001). No interprofessional difference was found in using patient information leaflets and point-of-care tests. CPs were highly likely to collaborate with GPs (p < 0.0001) but both professionals believed that policies that support GP–CP collaboration are needed to implement GPPAS intervention strategies. The collaboration challenges in implementing AMS were found at the level of persons, logistics, organisations, and policies.	There are opportunities for GP–CP collaboration in AMS, however, health system structures supporting routine collaboration and collaborative practice agreements between GP and pharmacy practices are key to foster GP/CP interprofessional trust for implementing AMS, developing AMS competencies together, and promomting communications for AMS activities.
**Study 6:** Systems Thinking Approach to Improve Antimicrobial Stewardship in Primary Care [25]	Rapid review	General practice and community pharmacy context	To analyse system thinking approaches how to improve implementation of AMS programmes in primary care involving interprofessional collaboration and communication	Systems thinking could help scrutinise the priority thoughts before, during, and after designing and implementing AMS programmes in primary care. Important areas involve how to incorporate AMS resources into an existing health system and by whom, understanding system structures and external factors that may operationalise AMS programmes, building interdependent AMS teams, and predicting a change in antimicrobial use over time. Opportunities are surmounting regarding how to transform antimicrobial-use behaviours culturally and structurally through establishing a sustainable AMS friendly health service model that ensures patient-centred but interprofessional antimicrobial care in primary care in the future.	Systems thinking approaches are important to determine the required arrangements for AMS resources, their access to health professionals, organisational system structures that would support routine AMS activities, dynamic system behaviours, and interprofessional communication and collaboration for practical design and implementation of AMS programmes in primary care.
**Study 7:** The Effectiveness of a Simple Antimicrobial Stewardship Intervention in General Practice in Australia: A Pilot Study [26]	Pilot implementation study in Australian primary care	A general practice in Victoria, Australia	To evaluate the impact of a novel GP educational intervention involving pharmacists improving appropriateness and guideline compliance of antimicrobial prescriptions	A GP educational AMS programme was significantly effective in improving appropriateness in antimicrobial selection (73.9% vs. 92.8%, RR = 1.26, 95% CI 1.18–1.34), duration (53.1% vs. 87.7%, RR = 1.65, 95% CI 1.49–1.83) and guideline compliance (42.2% vs. 58.5%, RR = 1.39, 95% CI 1.19–1.61).	The implementation of a GP educational programme involving pharmacist is effective to significantly improve appropriateness and guideline compliance of GPs’ antimicrobial prescriptions. The findings indicate that GP/pharmacist AMS education has an important role and should be sustainably continued for antimicrobial education within general practice.

**Table 2 antibiotics-11-01158-t002:** Key problems, facilitators, opportunities, and quality improvement strategies to routinely implement AMS activities through collaboration between GPs and CPs.

Key Problems	Facilitator	Opportunities	SEIPS 2.0 Components	Proposed Quality Improvement Strategies
Limited education, training, and professional development regarding AMS practice, strategies, and goals	The implementation of AMS training courses as part of the professional development modules of GPs and CPs are essential for their competency in AMS and participation in collaborative AMS activities. Incorporation of these courses into the GP/pharmacy curriculum of undergraduate and graduate programme would be valuable.	GPs (46.4% of 386) and CPs (76.5% of 613) felt that they would need AMS education and training [22,23]. Most GPs (72% of 386) and CPs (87.3% of 606) strongly agreed to receive AMS training in the future [24]. Their willingness is an opportunity to facilitate AMS training programme in general practices and community pharmacies in Australia	Person	**GPPAS implementation model for AMS educational programmes**
Potentially limited resources for routine use to make a shared decision about antimicrobial use during consultation and counselling with a patient	Antibiotic checklist, AWaRe tools, and patient-facing information leaflets	Less than 25% of surveyed GPs and CPs used patient information leaflets [22,23] and both professionals reported that these information sheets were not readily available for routine use [24]. They believed that these resources would help improve patient awareness and patient pressure on antibiotic use.	Physical environment Tools and Technology	**Access to AMS resources**
Patient expectation to receive an antimicrobial prescription while visiting a GP/CP with a symptom non-suggestive of antimicrobial use	Patient education and antimicrobial awareness campaigns, provide a patient with a take-home message about how to self-care and seek further advice in treating self-limiting infections	Most GPs (76.8% of 383) and CPs (82.4% of 608) educated patients about unintended consequences of antimicrobial use (e.g., AMR and gut effect) but without using any formal tools [22,23]. GP/pharmacy practice-based supply of these resources could be an opportunity to disseminate antibiotic awareness common messages at a community level through GP/pharmacist joint venture projects.	Person Task	**GP-patient communication**
				**CP-patient communication**
Lack of GP/CP communication regarding an antimicrobial prescription including a delayed antimicrobial prescription	Receptiveness of GPs to accept CPs’ recommendations Information technology support, telehealth-led antimicrobial reviews, case conferencing to discuss antimicrobial review outcomes GP-pharmacist partnerships to avoid early dispensing of delayed prescribed antimicrobial(s)	GPs felt that they should be receptive to CPs’ recommendations on the choice (50.5% of 381) and dose (63% of 382) of antimicrobials [22]. GPs’ willingness to accept that pharmacists’ recommendations where appropriate can be considered as pivotal for GP–CP collaboration with antimicrobial prescriptions.	Person Organisation Tools and technology Policy	**GPPAS implementation model for delayed antimicrobial prescriptions**
				**GPPAS implementation model for routine review of antimicrobial prescriptions **
No local or national GP/pharmacy practice agreements and policies supporting collaboration in AMS	GP/CP practice agreements and relevant policies	Most CPs (94% of 606) and the majority of GPs (60.9% of 381) believed that a policy supporting their collaboration in AMS was needed to improve AMS in the community [24]. Future policy initiatives surrounding the establishment of GP/pharmacy practice agreements could support implementation of GP/pharmacist collaborative AMS strategies at the local level in the short term and the national level in the long run.	Organisation External environment	**Policy environment**
Poor tracking and monitoring system to identify inappropriate prescribing and provide feedback	Validated tools supporting antimicrobial audits GP/pharmacist local AMS team.	GP-pharmacist collaborative antimicrobial audit models were interprofessionally (46.1% of GPs and 86.5% of CPs) supported to optimise antimicrobial therapy [24]. CPs’ optimistic desire to be involved in antimicrobial audit programmes can facilitate implementation of an audit feedback strategy in primary care.	Organisation	**GPPAS implementation model for antimicrobial audits and feedback programmes**
CPs’ lack of access to clinical indications and diagnostic reports to review antimicrobial prescriptions	User friendly “MY Health Records”, GPs’ mandatory reporting of clinical indication for an antimicrobial prescription	Most CPs believed that having access to a patient’s clinical and diagnostic information would assist reviewing the guideline adherence of prescribed antimicrobial(s).	Physical environment	**GPPAS implementation model for antimicrobial reviews **
				**Organisational system structure**
Lack of technology support to make optimal decisions about antimicrobials in a busy practice environment	CDSS, eTG (Antibiotic) integrated with prescribing and dispensing software	Nearly 30% of surveyed GPs and CPs [22,23] felt that, if AMS resources were linked with prescribing and dispensing software, considering AMS in a busy environment would be easier.	Tools and technology	**Access to resources**
Diagnostic uncertainty about the cause of infections	Availability and accessibility of point-of-care tests in GP and pharmacy facilities. Policy supporting the use of point-of-care diagnostic tests to determine the cause of patient infections, either bacterial or viral.	Less than 20% of surveyed GPs (N = 386) and CPs (N = 613) used point-of-care tests [24] but both professionals were supportive to introduce evidence-based point-of-care testing facilities to improve their diagnosis and antibiotic treatment decisions for patients seeking treatment for common and acute infections.	Tools and technology External environment	**GP/pharmacist diagnostic stewardship model**
				**Policy environment**

## Data Availability

All datasets collected and analysed are available to the corresponding author on reasonable request.

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
