# Peer review of "A Novel GPPAS Model: Guiding the Implementation of Antimicrobial Stewardship in Primary Care Utilising Collaboration between General Practitioners and Community Pharmacists"

_antibiotics, 2022, doi:10.3390/antibiotics11091158_

Round 1
Reviewer 1 Report
This manuscript describes and proposes an integrative, innovative and ambitious model for defining a collaborative between general practitioners and pharmacists to improve primary care antibiotic stewardship.
The following comments pertain:
· Whilst the content is of paramount importance for the reader, the manuscript is poorly structured and section and sub-section numbering requires major revision. The Reviewer is aware that Antibiotics now accepts free format submission and don’t have strict formatting requirements, but the authors are nevertheless advised, to perhaps consider conventional formatting.
· Numerous grammatical errors (some listed below) are evident
· Row 28. Five out of place
· Table 1.
o -Study 2. The following is out of place under Key Outcomes and should be moved to Methods:
“This study findings informed and guided the development of nationwide survey study to comprehensively understand AMS situation in Australian community pharmacy including current challenges”.
o -Study 6. The following sentence needs to be revised under Key Messages as it doesn’t currently make sense grammatically:
“There is an importance of systems thinking to figure out the optimal arrangement of AMS resources”.
· Table 2.
o The title is far too long
“Synthesised summary evidence regarding the key problems, facilitators, opportunities and quality improvement strategies for Australian GPs and CPs to rigorously involve, collaborate and routinely implement AMS activities that inform the design of the GPPAS model framework”
o 1st row under Opportunities
-Quoting a study here without reference is inappropriate. Please revise and summarize opportunities and not providing results per se
-“GPs (46.4% of 386) and CPs (76.5% of 613) felt that they would need AMS education and training. Most GPs (72% of 386) and CPs (87.3% of 606) strongly agreed to receive AMS training in future”
-This applies to all the studies quoted in this column
· Proposed quality improvement strategies
o The Reviewer is unsure what the meaning of colour coordinated strategies are meant to be as they don’t correlate to colours used in Figure 2
o Some of these in the column do not relate to quality improvement. Consider revising column heading
· Methods
o The reviewer is uncertain why Background, Methods, Results, Discussion, Conclusion don’t follow conventional flow as it is otherwise confusing. As a consequence, currently Figure 2 precedes Figure 1
· Supplemental Table 2
o What is the purpose of uploading the GUIDED checklist – a guideline for reporting intervention development studies? Where is it referred to in the manuscript? If it is, provide context please and explain page numbers in 2nd column
Besides the construction of the manuscript, the content is pivotal for readers to understand, how to inform a comprehensive multi-modal strategy to improve AMS in primary care. Utilizing the Systems Engineering Initiative in Patient Safety (SEIPS 2.0) model for ambulatory AMS is promising. However, as stated by the authors, the complex nature of AMS interventions necessitates future qualitative studies using grounded theory or behavioural approaches, to best refine the proposed model.
Author Response
Reviewer 1
This manuscript describes and proposes an integrative, innovative and ambitious model for defining a collaborative between general practitioners and pharmacists to improve primary care antibiotic stewardship.
The following comments pertain:
- Whilst the content is of paramount importance for the reader, the manuscript is poorly structured and section and sub-section numbering requires major revision. The Reviewer is aware that Antibiotics now accepts free format submission and don’t have strict formatting requirements, but the authors are nevertheless advised, to perhaps consider conventional formatting.
Thanks so much for highlighting the paramount importance of this work. The structure of the manuscript has been significantly improved by amendment of headings, subheadings, and numbering of section(s) and sub-sections. To improve the structure and flow of the text, results and discussion sections have been significantly improved as well. All amendments are found in track changes.
- Numerous grammatical errors (some listed below) are evident
We have addressed grammar issues after thorough review of the manuscript text.
- Row 28. Five out of place
Amended
- Table 1.
o -Study 2. The following is out of place under Key Outcomes and should be moved to Methods:
“This study findings informed and guided the development of nationwide survey study to comprehensively understand AMS situation in Australian community pharmacy including current challenges”.
Outcomes column has been revised with clarity
o -Study 6. The following sentence needs to be revised under Key Messages as it doesn’t currently make sense grammatically:
“There is an importance of systems thinking to figure out the optimal arrangement of AMS resources”.
The sentence has been revised.
- Table 2.
o The title is far too long
“Synthesised summary evidence regarding the key problems, facilitators, opportunities and quality improvement strategies for Australian GPs and CPs to rigorously involve, collaborate and routinely implement AMS activities that inform the design of the GPPAS model framework”
The title has been shortened with “Key problems, facilitators, opportunities and quality improvement strategies to routinely implement AMS activities by collaboration between GPs and CPs”
o 1st row under Opportunities
-Quoting a study here without reference is inappropriate. Please revise and summarize opportunities and not providing results per se
-“GPs (46.4% of 386) and CPs (76.5% of 613) felt that they would need AMS education and training. Most GPs (72% of 386) and CPs (87.3% of 606) strongly agreed to receive AMS training in future”
-This applies to all the studies quoted in this column
Relevant references have been incorporated and opportunities reported in each column have been revised and improved with explanation with summary languages.
-Proposed quality improvement strategies
o The Reviewer is unsure what the meaning of colour coordinated strategies are meant to be as they don’t correlate to colours used in Figure 2
- Some of these in the column do not relate to quality improvement. Consider revising column heading
The colour coordinated strategies have been corelated with the figure 1. The title and content of the table 2 have been amended.
-Methods. The reviewer is uncertain why Background, Methods, Results, Discussion, Conclusion don’t follow conventional flow as it is otherwise confusing. As a consequence, currently Figure 2 precedes Figure 1
“Antibiotics” journal follows method section to be reported at the end of other sections which is different from the conventional flow. Figure numbers have been corrected.
-Supplemental Table 2
o What is the purpose of uploading the GUIDED checklist – a guideline for reporting intervention development studies? Where is it referred to in the manuscript? If it is, provide context please and explain page numbers in 2nd column
The GUIDED checklist has been used for quality and consistency of reporting the GPPAS intervention development. The reference has been cited at the method section using reference 90.
-Besides the construction of the manuscript, the content is pivotal for readers to understand, how to inform a comprehensive multi-modal strategy to improve AMS in primary care. Utilizing the Systems Engineering Initiative in Patient Safety (SEIPS 2.0) model for ambulatory AMS is promising. However, as stated by the authors, the complex nature of AMS interventions necessitates future qualitative studies using grounded theory or behavioural approaches, to best refine the proposed model.
Thanks for highlighting the pivotal importance of the GP-pharmacist collaborative multimodal strategies to foster AMS in primary care. The refinement work as stated has been in progress.
Reviewer 2 Report
This manuscript entitled "A novel GPPAS model: guiding the implementation of antimicrobial stewardship in primary care utilizing collaboration between general practitioners and pharmacists"
The authors presented a seven-component exploratory study to inform General Practioner (GP)- pharmacist antimicrobial stewardship (GPPAS) model through Interprofessional collaboration.
They make a framework for AMS implementation from a series of their research in Australia and with reference to other countries.
Their model is informed by seven interactive domains demonstrating how AMS may be implemented in Australian primary care by fostering patient-centeredness and GP-pharmacist collaboration.
The seven interactive domains included
1) pharmacist-patient, 2) GP-patient, 3) GP-pharmacist,
4) resource access (tools and technology), 5) organizational structure, 6) task and 7) policy environment to foster interpersonal and interprofessional collaboration in AMS.
These results are informative enough and may help change antimicrobial use patterns and preserve the utility of antibiotics for tratement.
However, specific improvements are needed in the paper.
-Results should start with a bit of narrative rather than tables and figures.
-Check spacing for the title on line 96
- In figure 1, some captions need to be corrected example, slightly above Rapid review study six, what do you mean by "Secondary research providing insights on system thing"? probably meant "system thinking"?
- And to the right of the green box "Develop,ment of the prototype model," did you mean sub modeling rather than submodellin? Please check for the accuracy of all the captions inside the figure boxes!
- Line 310 The GP-CP collaborative routine task part needs more details
- Check how SEIPS is written and make it consistent on lines 80, 418, 428
- Lines 122-128, check sentence punctuations like the use of: and others
- Line 353 " You refer to "Our study for..." This is not a research article but rather a review. I think this manuscript is a review work and, throughout the text, should be referred to as such.
- Line 389-391 There is a need to tight in the methodology used for this study,
What is given in these lines "This was an exploratory study framed with..."
The later part discusses " a systematic review, a scoping review, nationwide surveys including qualitative components, a rapid review and a pilot study to inform a prototype GPPAS implementation model. The overarching project collecting secondary and primary evidence was conducted between 2017 and 2021."
- This later part discusses the previous studies; I think you should explain which methodology you used to collate all these studies to advise the design of this work.
-Please start with which review strategy/design was used to inform the methodology for your work on "A novel GPPAS model -framework."
- Line 419: replace: with a full stop
- Please change minor grammar issues throughout the text.
Author Response
Reviewer 2
This manuscript entitled "A novel GPPAS model: guiding the implementation of antimicrobial stewardship in primary care utilizing collaboration between general practitioners and pharmacists"
The authors presented a seven-component exploratory study to inform General Practioner (GP)- pharmacist antimicrobial stewardship (GPPAS) model through Interprofessional collaboration.They make a framework for AMS implementation from a series of their research in Australia and with reference to other countries. Their model is informed by seven interactive domains demonstrating how AMS may be implemented in Australian primary care by fostering patient-centeredness and GP-pharmacist collaboration. The seven interactive domains included 1) pharmacist-patient, 2) GP-patient, 3) GP-pharmacist, 4) resource access (tools and technology), 5) organizational structure, 6) task and 7) policy environment to foster interpersonal and interprofessional collaboration in AMS. These results are informative enough and may help change antimicrobial use patterns and preserve the utility of antibiotics for treatment. However, specific improvements are needed in the paper.
Highly appreciate your positive comments on the work and feedback for further quality improvements
-Results should start with a bit of narrative rather than tables and figures.
Results have been explained using subheadings and narrative details
-Check spacing for the title on line 96
Corrected
- In figure 1, some captions need to be corrected example, slightly above Rapid review study six, what do you mean by "Secondary research providing insights on system thing"? probably meant "system thinking"?
Corrected
- And to the right of the green box "Development of the prototype model," did you mean sub modeling rather than submodellin? Please check for the accuracy of all the captions inside the figure boxes!
Amended
- Line 310 The GP-CP collaborative routine task part needs more details
This component has been explained with more details
- Check how SEIPS is written and make it consistent on lines 80, 418, 428
Amended for consistency
- Lines 122-128, check sentence punctuations like the use of: and others
Amended
- Line 353 " You refer to "Our study for..." This is not a research article but rather a review. I think this manuscript is a review work and, throughout the text, should be referred to as such.
The GPPAS novel framework has been solely guided by the published studies of our research team. The work is a compilation of all our related studies as part of a PhD program. All the results of seven studies have been analysed and cumulative outcomes has built a foundation and structure to inform the prototype GPPAS model. International evidence has been used to support our evidence for the GPPAS sub-models and model components.
- Line 389-391 There is a need to tight in the methodology used for this study, What is given in these lines "This was an exploratory study framed with..." The later part discusses " a systematic review, a scoping review, nationwide surveys including qualitative components, a rapid review and a pilot study to inform a prototype GPPAS implementation model. The overarching project collecting secondary and primary evidence was conducted between 2017 and 2021." - This later part discusses the previous studies; I think you should explain which methodology you used to collate all these studies to advise the design of this work. -Please start with which review strategy/design was used to inform the methodology for your work on "A novel GPPAS model -framework."
Thanks for the opportunity to clarify the methodological details. The method section has been significantly amended to clarify and explain the methods applied to inform the GPPAS model framework. A critical synthesis approach was used to collect evidence from all our previous studies and framework analysis approach was used to collate evidence to determine model component of the GPPAS model using a theoretical framework of the SEIPS 2.0 model.
- Line 419: replace: with a full stop
Amended
- Please change minor grammar issues throughout the text.
We have addressed grammar issues after thorough review of the manuscript text. All amendments are found in track changes.